# "Mitigating cancer pain: What else matters?"— A qualitative study into the needs and concerns of cancer patients in Sri Lanka

N. P. Edirisinghe[1,2]*, P. T. R. Makuloluwa[3], A. A. T. D. Amarasekara[4], C. S. E. Goonewardena[5,6]

1 Faculty of Nursing, Department of Fundamental Nursing, University of Colombo, Colombo, Sri Lanka,
2 Faculty of Graduates Studies, University of Sri Jayewardenepura, Sri Jayewardenepura, Sri Lanka,
3 Faculty of Medicine, Department of Clinical Sciences, General Sir John Kotelawala Defence University, Dehiwala-Mount Lavinia, Sri Lanka, 4 Faculty of Allied Health Sciences, Department of Nursing and Midwifery, University of Sri Jayewardenepura, Sri Jayewardenepura, Sri Lanka, 5 Faculty of Medical Sciences, Department of Community Medicine, Cancer Research Center, University of Sri Jayewardenepura, Sri Jayewardenepura, Sri Lanka, 6 Faculty of Medical Sciences, Cancer Research Center, University of Sri Jayewardenepura, Sri Jayewardenepura, Sri Lanka

* keenirosha@yahoo.com

## Abstract

### Objectives

In Sri Lanka, cancer is a significant contributor to both morbidity and mortality rates. In 2022, 33,243 new cancer cases were reported, resulting in an age- standardized incidence rate of 106.9 per 100,000 individuals. The overall experience of cancer pain reflects patients' needs and concerns. Therefore, a thorough understanding of the patient's needs and concerns is crucial to implementing satisfactory pain outcomes. This study aims to explore the needs and concerns of patients with cancer pain in Sri Lanka.

### Methods

This study employed a descriptive qualitative approach among purposively selected patients with cancer and registered at the pain management unit. Participants recruited were 18 years or older with cancer-related pain. Noncancerous pain and those with psychological disorders, and brain metastases were excluded. Twenty-one semi-structured interviews were conducted until data saturation using a semi-structured interview guide, each lasting 30–60 minutes. Data were analyzed by Graneheim and Lundman's content analysis method.

### Results

The study primarily involved participants aged 51–60 Sinhalese Buddhists. It highlighted two main themes: 'Changes in normal lifestyle' and 'Needs and expectations'. The 'Changes in normal lifestyle' theme included subthemes like 'Functional limitations', 'Emotional reactions', 'altered interpersonal relationships', and 'Socio-financial problems'. The 'Needs and expectations' theme covered desires for a 'Pain-free life', a return to a 'Normal lifestyle', and

**Data Availability Statement:** All relevant data are within the manuscript and its Supporting Information files.

**Funding:** This study was funded by the Cancer Research Center, Faculty of Medical Sciences, University of Sri Jayewardenepura, Sri Lanka (Grant No: 002/2017).

**Competing interests:** The authors have declared that no competing interests exist.

the 'Need for a caregiver'. The findings emphasize that the most significant issue for cancer patients is the disruption to their normal lifestyle due to various challenges, while their primary need is to live without pain.

## Conclusions

'Life without pain' is a cancer sufferer's greatest need while 'changes in normal lifestyle' owing to bio-psycho-social-spiritual problems is their primary concern.

## Introduction

Pain is a major cause of cancer-related suffering. Therefore, a thorough understanding of patients' needs and concerns is crucial to implementing satisfactory pain outcomes. According to the World Health Organization (WHO), cancer killed almost 10 million people in 2020, or one in six [1]. In 2022, 33,243 new cancer cases were reported, resulting in an age-standardized incidence rate of 106.9 per 100,000 individuals. It accounted for 19145 deaths in the same year with an age- standardized mortality rate of 59.0, the growing burden of cancer in Sri Lanka underscores the urgent need to address cancer-related pain comprehensively [2].

Pain affects 66% of oncology patients [3] and affects all aspects of life. Cancer pain significantly impacts patients' emotions, cognitive performance, daily activities, and family and social interactions. A study found that 73.2% of cancer patients reported severe pain, with "normal works" and "sleep" being the most affected in Sri Lanka [4].

Cancer patients may live longer due to early detection and advancements in treatment modalities [5]. On the other hand, patients are subjected to uncomfortable diagnostic and therapeutic interventions, increasing the risk of experiencing pain. Untreated cancer pain is prevalent in Asia, at 59% compared to 40% in Europe and 39% in the USA, despite advances in cancer therapy and increasing access to supportive care [6]. A clinical audit found suboptimal pain assessment in cancer patients [7], nurses lack autonomy and specialized knowledge in cancer pain management, often working in a task-oriented system that neglects patients' pain needs [8], In Sri Lanka, limited healthcare resources and inadequate pain management facilities exacerbate these challenges. Pain is still widely misunderstood, under-reported, and frequently undertreated, resulting in significant distress for the patient and his or her close relatives.

Cancer-related pain can occur before diagnosis, during treatment, and at the end. Pain is common and may persist, return, or change. Long-term disease and therapeutic impacts affect even cured individuals. Advanced cancer patients may mistake increased pain for disease progression, creating silent fear, grief, and lower quality of life. Patients, especially cancer survivors, may also need unnecessary hospitalizations, outpatient visits, or long-term psychosocial treatment due to untreated or poorly controlled pain. Studies show that cancer pain is poorly controlled and that effective pain medicines are not used for optimal pain relief, despite the well-documented causes of poor pain therapy [9].

Insufficient pain assessment is the most common mistake in pain management, even though pain can be detected through self-reporting, physiological testing, and behavioral observation [10]. Cancer pain often causes biopsychosocial issues; hence many studies stress holistic care [11]. Cancer patients' participation in research helps them understand cancer pain. Their thoughts, experiences, and ideas emphasize life with cancer pain.

According to studies conducted in Brazil, China, Taiwan, Korea, and Turkey, pain has adverse effects on appetite, sleep, fatigue, daily activity, general appearance, mood, family support, financial status, walking ability, relationships with others, enjoyment of life, nutrition, mobility, emotional status, and overall QoL of patients [12–16]. In a study among 5084 adult patients, 69% reported pain-related difficulties with everyday activities [17]. Patients with pain have distinct demands, such as pain alleviation, reassurance from their healthcare provider, an acceptable degree of functioning, and the capacity to live a normal lifestyle. However, it is important to realize that everyone reacts to pain differently depending on their personality, pain threshold, expectations, and requirements. Therefore, pain management must also be individualized [18]. Further, everyone needs to find solutions to the problems mentioned.

The primary objectives of a cancer-care system are to reduce cancer-related mortality and enhance the quality of life (QoL) of survivors. A critical aspect of improving QoL for cancer patients is the effective recognition and management of pain, which significantly impacts their physical, emotional, and social well-being. In Sri Lanka, a low-middle-income country, the healthcare system faces substantial challenges in adequately recognizing and addressing cancer pain and its associated phenomena. This gap in care exacerbates the burden on patients, their families, and society at large.

Despite advancements in medical therapies, there remains a lack of comprehensive understanding of the essential needs and concerns related to cancer pain, particularly in resource-constrained settings like Sri Lanka. The country's healthcare system, while acknowledging the importance of holistic care, struggles with significant gaps in the implementation of such approaches. This is especially crucial given Sri Lanka's ongoing demographic and epidemiological transition, characterized by an aging population and an increasing prevalence of non-communicable diseases [19].

The study aims to address the challenges associated with cancer pain in Sri Lanka by deepening the understanding of the substantial needs and issues faced by patients. Beyond focusing solely on pain relief, this research will investigate the broader psychosocial concerns that significantly impact the quality of life for cancer patients. These include emotional, psychological, cultural, and social factors that influence patients' experiences and coping mechanisms. By examining these areas, the study seeks to enhance pain recognition and contribute to the development of more effective, patient-centered interventions, ultimately aiming to reduce the overall burden of cancer pain in the country.

In light of these factors, it is vital to emphasize the significance of conducting this study using a qualitative research approach. Qualitative methods provide unique insights into patients' lived experiences, perspectives, and unmet needs, offering a rich understanding of the complexities surrounding cancer pain. By capturing these nuances, the study seeks to identify patient needs and problems of effective pain management and to understand the emotional and psychological impacts of cancer pain in the Sri Lankan context. Ultimately, these insights will inform the development of more effective, patient-centered interventions tailored to the unique needs of cancer patients in Sri Lanka.

## Methods

### Study design

A qualitative descriptive study was conducted to delve into the lived experiences of individual needs and major concerns of patients with cancer pain in Sri Lanka. This study adopts a constructivist perspective, acknowledging the subjective nature of experiencing cancer pain and seeking to explain and comprehend the many interpretations attributed to these experiences by individuals. Data was collected from November 2018 to April 2019.

## Study setting and participants

Apeksha Hospital, also known as the National Cancer Institute, was selected as the study setting due to its status as the leading tertiary care center for oncology in Sri Lanka, providing comprehensive cancer services to a diverse and extensive patient population. The hospital's specialized oncology units, pain management services, and multidisciplinary healthcare workers enable a holistic exploration of cancer pain across various patient demographics. Additionally, its central role in the nation's cancer care system ensures that the study's findings will be relevant and impactful in shaping future cancer pain management strategies in Sri Lanka.

Participants for this study were purposefully selected from the Pain Management Unit at Apeksha Hospital to capture a wide range of experiences with cancer pain. The study included patients with both curable cancer and those at advanced or palliative stages, ensuring a comprehensive exploration of pain and its management across different phases of the cancer journey. This selection process was designed to reflect the diverse interpretations of pain and the varying needs associated with each stage, thereby providing a richer understanding of the challenges faced by cancer patients in Sri Lanka.

The inclusion criteria for study participants were; patients aged 18 and older with cancer-related pain (a pain score of three or more on a Numerical Rating Scale) who attended the Pain Management Unit at Apeksha Hospital in Maharagama. All participants were required to understand and speak Sinhala and provide informed written consent. Patients were excluded if they experienced non-cancerous pain, were frail or mentally unfit, or were disoriented with evidence of brain metastases. Purposive sampling was used to select patients from various age groups, sexes, educational levels, cancer types, and treatment modalities, allowing for a broad representation of experiences within the study.

## Study instruments

The interviews followed a semi-structured interview guide (S1 File). The guide's concepts and questions were based on a comprehensive literature review and revised by subject specialists. The semi-structured interview guide in this study was designed to comprehensively explore participants' experiences with cancer pain. It included a mix of open-ended questions and probing questions to encourage deeper discussion. The interview typically began with a broad question, such as "Can you tell me about your experience with cancer pain?". This question was intended to set the stage for the discussion and to give participants the freedom to highlight the aspects of their pain experience that were most significant to them. Probing questions were employed as needed throughout the interview to delve deeper into topics raised by the participants. For instance, if a participant discussed a specific problem they encountered, the interviewer could ask, "Can you explain more about how this problem affects your daily life?" This approach ensured that the interviews remained flexible and responsive to the unique experiences of each participant while maintaining a focus on the key research questions.

The semi-structured guide was pre-tested on five in-patients with cancer pain. This helped identify difficult probing questions and determine if the interviewer guide received the desired range of responses. The timing of the interviews was determined, and questions about transcription were also clarified. The guide was revised after the pre-test.

## Data collection

Administrative clearance was obtained from the Director of Apeksha Hospital, Maharagama, prior to initiating the data collection process. The principal investigator (PI), conducted semi-structured interviews (n = 21) until theoretical data saturation was attained with the use of an interview guide. The PI's background in nursing and qualitative research, combined with her

previous experience in conducting in-depth interviews and effective communication, ensured the rigorous collection of data and enhanced the reliability of the findings. The interviewer introduced herself and described the purpose of the interview. The informant was assured that the information would be treated with confidentiality in a way that his/her name would never be identified in any notes taken and written reports. It was also stressed that he/she was allowed to express any opinion and that no judgment would be made. Care was taken to avoid using the word 'cancer' or any other word with a similar meaning to minimize the distress for the informant. The phrase 'the disease treated at this hospital' was used instead. The informant acknowledged that clinical staff would not be informed until indicated. In such cases, the informant's consent was sought. Informed written consent, including recording the interview, was obtained from each participant before the interview.

The interviews were conducted in a quiet room at the pain clinic or in a separate area in the ward where the participants felt comfortable, and their privacy was ensured. Nonverbal cues were noted. Throughout the interview, the interviewer was open and nonjudgmental. Finally, the interviewer summarized the conversation and assessed the respondent's validity. The interviews lasted 30–60 minutes.

## Data analysis

The qualitative data were analyzed using Graneheim and Lundman's content analysis method [20]. The process began with the audio-recording of all interviews, which were then transcribed verbatim by the interviewer in the original language (Sinhala). To ensure the accuracy and completeness of the data, each transcription was meticulously checked against the audio recordings. The transcripts were then translated into English by a bilingual expert familiar with the research context, ensuring that the nuances and meanings of the original language were preserved (S2 File).

The coding process was data-driven, with codes emerging directly from the interview data rather than being based on pre-identified themes. After familiarization with the data through repeated readings of the transcripts, meaningful units of text were identified and highlighted. These units included phrases, sentences, or paragraphs that were relevant to the research questions. Initial codes were assigned to these meaningful units, and similar codes were grouped into broader categories. The initial coding was conducted by the principal investigator, and these initial codes were reviewed by the research team. The team included qualitative research experts who contributed to refining the coding framework. To ensure consistency and accuracy, the codes were discussed in team meetings where the researchers collaboratively reviewed and agreed upon the coding of the raw data. In cases where disagreements arose, these were resolved through discussion until a consensus was reached. The analysis was iterative, allowing for the integration of new data as additional interviews were conducted. This approach helped in refining categories and identifying new themes. The categories were further broken down into subcategories to capture more specific aspects of the data. Finally, overarching themes were identified by analyzing the categories and subcategories, representing the central ideas and latent meanings within the data. These themes were interpreted and refined through ongoing discussions among the research team until a consensus was reached. During the interviews, nonverbal responses such as facial expressions, gestures, body language, and tone of voice were carefully observed by the interviewer and noted in field notes. these nonverbal cues were used to enrich the interpretation of the data, helping to identify underlying emotions or concerns that might not have been explicitly verbalized by the participants. This added a layer of depth to the themes related to pain, emotional distress, and coping mechanisms.

## Trustworthiness

The trustworthiness of this study was ensured through the application of Guba and Lincoln's guidelines, which emphasize four key components: credibility, transferability, dependability, and confirmability [21]. To achieve credibility, several strategies were employed. The principal investigator (PI) spent considerable time in the field, building strong relationships with participants, which facilitated trust and led to the collection of rich, in-depth data. To validate the findings, two cancer pain patients, who were not part of the study, reviewed the codes and themes to ensure they accurately represented patient experiences. Additionally, interviews were conducted with participants from diverse socio-demographic backgrounds to ensure a comprehensive representation. Two external qualitative research experts further reviewed the units of meaning, codes, subcategories, categories, and themes, providing feedback that was incorporated into the final analysis.

Transferability was enhanced by purposefully selecting a diverse sample of participants, encompassing a wide range of experiences with cancer pain. This diversity increases the likelihood that the findings can be applied to similar settings, making the results more generalizable.

Dependability was ensured by maintaining detailed documentation of the research process, including data collection and analysis procedures. This comprehensive record allows other researchers to follow the steps taken and verify the study's findings. Furthermore, multiple transcripts were reviewed and coded by the PI, with the codes compared and refined to ensure consistency and reliability.

To achieve confirmability, the PI maintained a reflexive journal throughout the research process, documenting personal reflections, potential biases, and decisions made. This reflexive practice, along with the use of field notes, helped ensure that the findings were grounded in the data rather than influenced by the researcher's preconceptions. The systematic documentation of these practices strengthened the confirmability of the study's results.

## Ethical considerations

Ethical approval was granted for this study by the Ethics Review Committee (ERC), Faculty of Medical Sciences, University of Sri Jayewardenepura, Sri Lanka (App No: 32/17). Comprehensive details about the study's aims, procedures, and significance were shared with the participants before gathering data. All participants gave their informed written consent, highlighting the voluntary nature of their involvement, their freedom to exit the study at any point without any consequences, and the guarantee that their information would remain confidential. Rigorous precautions were taken to maintain participant privacy and anonymity. Personal or identifiable information obtained during the research was treated with the utmost confidentiality, securely stored, and made available only to the investigators involved in the study.

## Results

Among the participants (n = 21), the majority were Sinhalese and Buddhists, with most falling within the 41–70 age group. Twelve (57.1%) were females, and 16 (76.2%) were married. Most participants had a monthly income of less than LKR.5000, and roughly 47% had completed Grades 6–11. Among the individuals, 57% reported pain for three months or more, and 52% had cancer for less than a year (Table 1). During data analysis, two primary themes emerged about cancer patients' needs and concerns. Each theme contained sub-themes with meaning units.

**Table 1.  Demographic characteristics of the study participants (n = 21).**

| Socio-demographic characteristics | | Frequency | Percentage % |
|---|---|---|---|
| Age (in years) | 18–30 | 2 | 9.5 |
| | 31–40 | 3 | 14.3 |
| | 41–50 | 5 | 23.8 |
| | 51–60 | 7 | 33.3 |
| | 61–70 | 4 | 19.0 |
| Ethnicity | Sinhala | 18 | 85.7 |
| | Tamil | 2 | 9.5 |
| | Muslim | 1 | 4.8 |
| Religion | Buddhism | 14 | 66.7 |
| | Catholic /Christianity | 4 | 19.0 |
| | Hindu | 2 | 9.5 |
| | Islam | 1 | 4.8 |
| Gender | Female | 12 | 57.1 |
| | Male | 9 | 42.9 |
| Current Marital status | Married | 16 | 76.2 |
| | Unmarried | 2 | 9.5 |
| | Divorced/ separated | 1 | 4.8 |
| | Widow | 2 | 9.5 |
| Highest level of education | Not been to school | 0 | 0 |
| | Grade 1–5 | 4 | 19.0 |
| | Grade 6–11 | 10 | 47.6 |
| | Grade 12–13 | 6 | 28.6 |
| | Graduate | 1 | 4.8 |
| | Postgraduate | 0 | 0 |
| Monthly income (Rs) | < 5000 | 18 | 85.7 |
| | 5001–10000 | 0 | 0 |
| | 10001–15000 | 0 | 0 |
| | 15001–20000 | 0 | 0 |
| | >20000 | 3 | 14.3 |
| Occupation | Yes | 4 | 19.0 |
| | No | 17 | 81.0 |
| Type of family | Nuclear | 11 | 52.4 |
| | Extended | 10 | 47.6 |
| Availability of helper | Yes | 15 | 71.4 |
| | No | 6 | 28.6 |
| Family responsibilities | Yes | 13 | 61.9 |
| | No | 8 | 38.1 |
| Time since diagnosis of cancer | < 1 year | 11 | 52.4 |
| | 2–3 years | 03 | 14.3 |
| | > 3 year | 07 | 33.3 |
| Type of Cancer | Uro-genital | 4 | 19.0 |
| | Gastro-intestinal | 6 | 28.6 |
| | Breast | 4 | 19.0 |
| | Other | 5 | 23.8 |
| | Lung | 2 | 9.5 |
| Duration of cancer pain | 1–3 months | 9 | 42.9 |
| | >3 months | 12 | 57.1 |

*(Continued)*

**Table 1.** (Continued)

| Socio-demographic characteristics | | Frequency | Percentage % |
|---|---|---|---|
| Presence of co-morbid diseases | Yes | 6 | 28.6 |
| | No | 15 | 71.4 |

## Theme 1: Changing normal lifestyle

Advanced cancer patients' daily lives are affected by pain. The subthemes were "functional limitations," "emotional reactions," "altered interpersonal relationships," and "socio-financial problems." Cancer-related pain affected all respondents. Physical, social, economic, and psychological distress were noted by patients. They regretted their pain, which left them depressed and hopeless most days. Table 2 lists the meaning units, codes, and categories for Theme 1: Changing normal lifestyle.

**Functional limitations.** According to participants, cancer pain had a negative impact on their everyday activities. As their condition advanced, they could not adjust their lifestyle to obtain better pain relief. Almost every patient stated that cancer pain had a detrimental impact

**Table 2. Theme 1: Changing normal lifestyle.**

| Meaning units | Condensed meaning unit | Codes | Categories | Themes |
|---|---|---|---|---|
| *"I cannot do housework, and I did all the housework. They are all ruined now."* | Patients could not attend to their self-care activities/other general work; pain interferes with all activities. | • Unable to do normal daily activities | **Functional limitations** | **Changing normal lifestyle** |
| *"I cannot do anything when the pain comes."* | Pain causes many discomforts (e.g., swallowing, walking) | • Physical discomforts | | |
| *"I cannot sleep when I have pain, which interferes with sleep.* | Pain interferes with sleep, interrupts sleep, or reduced sleep quality | • Disturbed sleep | | |
| *"I cannot cook now. I do not eat the way I want to. . ."* | Patients were unable to do their daily activities as they wish | • Limited freedom of life | | |
| *"I cry most of the time. I tell the people who talk with me that this is my grief. . . . . ."* | They feel sad due to the non-fulfillment of their hopes or expectations due to pain | • Feeling disappointed | **Emotional reactions** | |
| *"I jumped off the roof once. I cannot bear the pain. . . . . . . What to do when the pain increases?"* | Having thoughts of dying as it is better than living with unbearable pain. Insecurity with pain. | Catastrophizing thoughts | | |
| *"I forget what I need to do. . . . I do not feel like doing anything.."* | Patients were unable to focus on a task or easily distracted due to pain | • Poor concentration | | |
| *"I feel angry when someone talks when I am in pain."* | Pain makes them feel annoyed or agitated easily. | • irritability | | |
| *"Now the pain really got worse and worse. It looks like I cannot wait any longer."* | Pain is becoming unbearable and worrying. The feeling of uncertainty and fear about the future | • anxiety | | |
| *"My life has changed more than ever. . . . What was done before cannot be done now."* | The patient was unable to attend to his/her previous roles in the family/society/workplace due to pain | • altered personality (role) | | |
| *"Social relationships and social works have almost stopped."* | Confined to the home as mobility aggravates the pain | • Interrupted social relationships | **Altered interpersonal relationships** | |
| *"I cannot do that (cooking) for my husband. I cannot do anything for him. I feel sad. . . ."* | Feeling of dejection with the inability to complete own role in the family | • Interrupted family/marital relationships | | |
| *"I do not have anybody to be with me. I have to pay Rs. 2000 per day to keep the bystander."* | Lack of family and social support. Additional expenses for treatments, traveling, investigations, and caregivers. Being in debt | Lack of family support Increased health cost | **Socio-Financial difficulties** | |
| *"Driving is my job. I cannot do it now. The job has now stopped."* | Inability to do the job due to pain | Loss of income | | |

on their functional level, daily activities, and sleep. They were unable to eat, walk, or cook. The pain made sitting, standing, or rolling incredibly uncomfortable.

"*I cannot do housework, and I cannot sleep; I did all the housework. They are all ruined now*". (P-01)

It was said that the lack of trigger and energy made mobility difficult. Fatigue led to a sedentary lifestyle. For example, numerous patients reported losing interest in or taking longer to complete chores in general.

"*Because of this pain, I cannot do anything. . . I cannot do anything when the pain comes. I get bored easily of everything.*" (P-14)

Sleep deprivation is another significant issue connected with cancer pain. Cancer pain results in poor quality sleep that is shorter in duration and interrupted frequently.

"*I cannot sleep when I have pain, and it interferes with sleep. I fall asleep when the pain goes away, and it may be daytime or night. . . If there is pain, I will never fall asleep. . .*". (P-07)

**Emotional reactions.** Almost all reported emotional alterations. Tearfulness, depression, anxiety, anger, and irritation were feelings not experienced previously. Many psychological issues were directly or indirectly linked to pain and disease. Pain has made some feel unhappy, scared, helpless, and hopeless.

"*I cannot do anything now. . . Because of this leg [showing her leg]. I am in a very miserable situation now. . .. Worried . . .. I am very sad. . .. (Crying). . .. I cry most of the time. . .*" (P-01)

Some individuals stated they could not concentrate and forget easily when the pain arises.

"*Sometimes, I get irritated with pain. Other than running here and there, I forget what I need to do. . .. I do not feel like doing anything*". (P-15)

Most individuals feel angry as the pain rises. They do not like to talk with anybody when the pain increases, and they are worried about their life thinking about how it was past and present.

"*I feel angry when someone talks when I am in pain. I have a concussion, and I cannot sleep. Annoying, no consolation. It is heartbreaking to see how I lived for so long and living now. How it was then, how it is now. . .*" (P-18)

Patients in this study reported fatalistic thoughts that there was nothing they could do to escape cancer pain or death or that they would prefer to die to be free of pain. They tend to cry and think repeatedly, and some have had suicidal ideations.

"*Now the pain really got worse and worse. It looks like I cannot wait any longer. Yes, I jumped off the roof once. I cannot bear the pain. . . That is so hard.*" (P-02)

**Interrupted interpersonal and social relationships.** The pain has affected most participants' relationships with spouses, relatives, and outsiders. Some respondents valued the aid of

friends, family, neighbors, and the workplace. Many described how important it was to maintain cherished relationships and how difficult it was when pain interfered, whether due to the need for physical assistance or seeing their carers distressed:

"*There are times when I get angry. I only see her [*wife*] tears when I scold her. I feel so sorry for her. . . so there is nothing to do, so much pain. . . then I say sorry. However, although I apologize, she might have been hurt a lot. We have been together with our family for 25 years now. She knows my helplessness. The heart feels that inner pain. So such incidents happen, I apologize to God, 'God forgives me, I have done something like this from my mouth to my wife.*" (P-16)*

Some patients fear their sickness. Pain prevents them from visiting anyone. Due to their discomfort, some informants feared socializing. Below are a few statements by them.

"*I do not even want anybody to visit me. If I cannot treat or care and not talk to them. . .*"

"*I do not like to go to any social events, temple, or wedding ceremonies since I cannot be bemoaning there. I feel my sight might be an unlucky sign. So, if I cannot be there without a mess, I do not want to go to such places. . ..*" (P-18)

**Financial difficulties.**   Pain has affected the lives of most of the participants. Most respondents listed money issues as a major source of concern. The most expensive charges were transportation to the hospital, medicines, and investigations. Religious activities and alternative remedies were also high costs. The issues of high expenses were exaggerated by the poor income generation resulting from the disease, which led to borrowing money.

Sometimes family members have to leave work to care for their loved ones. If the victim is the family's breadwinner, they faced too many problems.

"*Driving is my job. I cannot do it now. If there is no pain, I can do my job, which is unbearable. I always feel pain. The only problem is the pain*" (P-02)

Most are incapacitated and need someone to care for them. In the absence of a family member, it was required to hire paid caregivers.

"*We do not have anybody to be with me* [as a caregiver at the hospital]. *I have to pay Rs. 2000 (2000LKR) per day to keep a caregiver. From where can I get so much money. . .? Rs10,000 (10000LKR) if I stay for five days. . .. [silence for a while]. . . My younger daughter was about to marry this year. The boy who was about to marry my daughter stayed with me at the hospital for a month, and finally, he lost his job. . .*" (P-16)

A woman with lung cancer from a low-income family revealed how she helped her family earn money while she was healthy. However, she is worried because she is not contributing now.

"*I used to sew carpets. I cannot do it now; Earlier, I was earning something too. However, now I have lost that too. It costs a lot of money for kids to learn. Now my husband has to do everything. So there are many money matters at home*". (P-14)

## Theme 2: Expectations and needs

Most participants are unsatisfied with their pain relief and live with pain. Meeting patients' expectations will undoubtedly result in increased patient satisfaction with care. The challenge was to extract genuine expectations and needs from individuals. Table 3 lists the meaning units, codes, and categories for Theme 2: Expectations and needs.

**Need for a caregiver.** Many patients felt that the primary caregiver's job is vital. When a housewife got sick, the other household members had to care for the patient and the housework. Thus, some of them could not continue in their usual jobs.

*"Mostly, I am alone at home. So, most of the time, there is no one to talk to. My wife looks after me 500% at home. . . but she cannot lift me. She is not strong enough to lift me. She is somewhat old now too. . .. she has lots of other household work to do as well. . .." (P-16)*

**Spend a normal lifestyle.** Participants expressed a desire to 'live normally' and attend to 'daily duties,' such as domestic chores, hobbies, and maintaining relationships with family and friends. These requests emphasized how pain was affecting them functionally, their ability to perform tasks, and in attending to their needs rather than their general health and function. All participants noted the importance of having control of their lives to ensure independent status without burdening the next of kin. Most participants explained what they had lost of their former selves. Participants described the need for a 'pain-free status', enabling them to resume their former lives. Many reported how their lives and everyday activities were kept on pause until pain relief was obtained and how they had lost pieces of their previous selves as a result of the pain.

*"Feeling we have stopped at the same place in our lives these days . . . because of this pain and disease. . . All the plans were stuck for a while. The essential thing is a full recovery." (P-11)*

**Pain-free life.** Participants expressed their need for mental and physical well-being with satisfactory pain relief. They expect it to happen either by completely curing the disease, providing satisfactory pain relief or through spiritual means. A significant finding was that participants preferred to present themselves as patients with cancer rather than dying from pain.

Table 3. Theme 2: Expectations and needs.

| Meaning units | Condensed meaning units | Codes | Categories | Themes |
|---|---|---|---|---|
| *"The only thing I want is to be free of pain. Nothing else is needed. It does not matter even if dying from cancer".* | Most of them need to live without pain, which bothers them more than cancer. | Need to live without pain | **Pain-free life** | **Expectations and needs** |
| *"God is the one who gave me this pain. I am always telling God. . . Please get this back."* | They believe that it is God who has given this pain and expects God to relieve pain. | The spiritual belief that God has the power to relieve pain | | |
| *"I do not want billions from somewhere . . .. I want to go out as before. . . That is all. . ."* | To live a life without a burden on the others | Need to be independent | **Spend a normal lifestyle** | |
| *"We do not have a bathroom inside, and Now my wife also cannot take me. If I have a wheelchair, it is easy for me to go out. . ."* | Need for physical aids to ease daily living | | | |
| *"I need someone to be with me. There is no one to give me anything even to eat and drink."* | Need somebody to help in self-caring | Dependency on others fulfilling personal care needs, security, and communication needs | **Need of a caregiver** | |
| *"I am talking to this one [bystander]. I forget all the mental problems in my head. . .."* | Need somebody to talk to distract from mental agony | | | |

"*The only thing I need is to live without pain... Nothing else is needed... It does not matter if I am dying of cancer ... I know that I have to die at some point... What I want is to die without being in pain.*" (P-15)

"*I have everything... I do not want anything. I do not have the temptation on anything. No value of anything to me... I want an immediate cure and get rid of this pain.*" (P-21)

## Discussion

Experiencing pain on top of cancer is overwhelming to the patient and the family. While pain management has been recognized as a critical aspect of cancer care globally, research exploring the specific concerns and needs of cancer patients in Sri Lanka has been limited. This study fills a gap in the literature by providing insights into how Sri Lankan patients experience cancer pain and its broader psychosocial impact. The findings contribute to the growing body of international research on cancer pain management, emphasizing the importance of addressing not only the physical aspects of pain but also the emotional, social, and cultural factors that influence patients' overall well-being. This study provided 21 cancer patients with a voice to discuss their experiences with illness and pain, deepening our understanding. The interview transcripts highlighted a lifestyle change and expectations/needs. The findings reflect how previous life experiences were embedded and interlaced with pain and participants' daily activities.

A major theme of this current study was changing normal lifestyles in terms of physical, social, and psychological elements. The impact of cancer pain on daily life and activities was repeatedly mentioned. However, similar losses were also addressed in relation to cancer, its treatment (chemotherapy), and cancer-related symptoms such as weakness and fatigue [22,23]. A complicated factor was the existence of comorbidities, which contributed to the loss of function. A recent Chinese study found "functional constraints" as a major theme and "daily living activities," "social communication," and "work" as sub-themes, similar to ours [23].

The current study participants' well-being was affected by their inability to undertake "normal" activities like shopping, walking in the garden, or socializing. The cancer literature shows that pain hinders independence and daily activities [24]. Social rejection, sadness, and loss of self were described by some cancer patients who lost mobility. They valued family and societal duties and felt less worthwhile without them. However, patients worry about being seen as a burden to their families, not only because of the disease's costs but also because of the care they receive, which impacts perception and powerlessness. Duggleby (2000) found pain adaptation difficult in 11 participants, 65-year-olds [25]. The pain increased their powerlessness and sorrow [26]. Thome and colleagues found that cancer pain was linked to death and dying fears in ten Swedish older adults [27]. Other research found similar worries about pain, illness progression, and pain treatments [28].

Participants in the current study ranked 'living a pain-free life,' "returning to a normal lifestyle,' and 'need for a caregiver' as their most essential expectations and needs. Survival and pain alleviation were the primary concerns of the participants. Analgesia was the primary concern. The people who participated in the study said pain control was necessary for their well-being. Xu and colleagues [23] reported pain relief as a top priority among 12 participants with cancer pain in China.

Patients resort to religion/spirituality for hope, confidence, and health. Research shows that religion and palliative care are generally beneficial. Religion and spirituality reduce stress and sadness and improve quality of life [26]. Spiritual care helps patients have hope (88%) and comfort (83%), according to research in Spain. Bovero et al stated that the spirituality is

positively associated with quality of life in terminal cancer patients [29]. Participants in Brockopp et al.'s [28] study acknowledged the importance of a supportive relationship with their caregivers.

One study examined patients' descriptions of a day when they thought their pain was 'adequately managed,' including interviewing 15 people with metastatic disease [30]. The study identified aspects that patients believe are critical for an 'acceptable day,' such as engaging in meaningful activities, relieving pain, and feeling well enough to socialize. McPherson and colleagues [22] found that several participants hesitated to discuss their pain with doctors or family members. Participants in this study expressed their desire for independence, including adaptation and dependency, pain, exhaustion, daily routines, sadness, and loss. Cancer reduces their freedom substantially. Mobility, domestic tasks, personal care, and social and recreational activities were described as realistic or under control. Patients sought normalcy through caring for or being with family.

The findings of this study revealed that demographic factors such as age, gender, and cultural background played a significant role in shaping the cancer pain experiences of the participants. For instance, the majority of the participants were Sinhalese Buddhists, and many turned to religion and spirituality as important coping mechanisms. Age also appeared to influence the participants' pain management strategies and perceptions. Participants in the 51–60 age group, many of whom were in advanced stages of cancer, frequently mentioned the physical limitations imposed by pain, such as the inability to perform daily activities like cooking or walking. Socioeconomic factors also impacted participants' access to healthcare and pain management resources. Patients from lower-income backgrounds expressed concerns about the financial burden of cancer treatment, which added to their emotional stress and sense of helplessness. This mirrors global findings on the socioeconomic disparities in cancer care, where limited financial resources often hinder patients' access to effective pain management.

The validity and reliability of research are directly related to the quality or trustworthiness of the results. In the context of this study, following strategies were used to enhance the study's trustworthiness. These strategies involve engaging participants, providing detailed descriptions, and working in the field for an extended period. While qualitative research does not seek to generalize findings to a larger population, it usually offers sufficient descriptive detail to enable the reader to assess whether the findings apply in other contexts. Given that saturation was achieved across a broad population, it is reasonable to assume that the findings apply to cancer patients treated in a similar setting.

The findings of this study have important implications for cancer pain management in Sri Lanka. Given the multidimensional nature of cancer pain, a more comprehensive, patient-centered approach to pain management is essential. The study underscores the need for healthcare practitioners to consider not only the physical symptoms of pain but also the psychological, social, and cultural factors that influence patients' well-being. Developing multidisciplinary care models that integrate medical, psychological, and spiritual support could improve the quality of life for cancer patients. Additionally, training healthcare professionals in culturally sensitive communication and pain management practices would be beneficial, particularly in addressing patients' emotional and spiritual needs.

Further research is needed to evaluate the effectiveness of individualized, multidisciplinary pain management protocols that address the unique cultural and psychosocial needs of cancer patients in Sri Lanka. Future studies could focus on the development and testing of such models, as well as their impact on patient outcomes. Additionally, research exploring healthcare professionals' attitudes and knowledge about pain management in low-resource settings like Sri Lanka could inform targeted interventions for improving care. Longitudinal studies could

also provide insights into how pain management needs to evolve throughout the cancer journey, particularly at end-of-life stages.

## Conclusion

Our study's findings indicate a variety of concerns and needs of patients experiencing cancer pain, which may not be addressed during a typical pain session. 'Pain-free' living was identified as the most essential need of patients with cancer pain, while 'change in normal lifestyle' due to adverse impact on 'bio-psycho-social-spiritual wellbeing' was recognized as the most pressing concern.

## Supporting information

**S1 File. Interview guide.**
(DOCX)

**S2 File. Participant supporting evidence.**
(XLSX)

## Acknowledgments

The authors, with great respect, appreciate the cooperation extended by the staff of Apeksha Hospital, Maharagama, Sri Lanka, and all the study participants.

## Author Contributions

**Conceptualization:** N. P. Edirisinghe, P. T. R. Makuloluwa, A. A. T. D. Amarasekara, C. S. E. Goonewardena.

**Data curation:** N. P. Edirisinghe.

**Formal analysis:** N. P. Edirisinghe, A. A. T. D. Amarasekara, C. S. E. Goonewardena.

**Funding acquisition:** C. S. E. Goonewardena.

**Investigation:** N. P. Edirisinghe, C. S. E. Goonewardena.

**Methodology:** N. P. Edirisinghe, P. T. R. Makuloluwa, A. A. T. D. Amarasekara, C. S. E. Goonewardena.

**Project administration:** N. P. Edirisinghe.

**Resources:** N. P. Edirisinghe, A. A. T. D. Amarasekara, C. S. E. Goonewardena.

**Supervision:** P. T. R. Makuloluwa, A. A. T. D. Amarasekara, C. S. E. Goonewardena.

**Validation:** N. P. Edirisinghe, P. T. R. Makuloluwa, A. A. T. D. Amarasekara, C. S. E. Goonewardena.

**Writing – original draft:** N. P. Edirisinghe.

**Writing – review & editing:** N. P. Edirisinghe, P. T. R. Makuloluwa, A. A. T. D. Amarasekara, C. S. E. Goonewardena.

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
