## [Decision Letter · Decision Letter 0]

12 Aug 2024

PONE-D-24-13898‘‘Mitigating Cancer Pain: What else Matters?”—A Qualitative Study into the Needs and Concerns of Cancer Patients in Sri LankaPLOS ONE

Dear Dr. Edirisinghe,

Thank you for submitting your manuscript to PLOS ONE. After careful consideration, we feel that it has merit but does not fully meet PLOS ONE’s publication criteria as it currently stands. Therefore, we invite you to submit a revised version of the manuscript that addresses the points raised during the review process.

We look forward to receiving your revised manuscript.

Kind regards,

Surangi Jayakody, MBBS, MSc, MD

Academic Editor

PLOS ONE

Journal Requirements:

“This study was funded by the Cancer Research Center, Faculty of Medical Sciences, University of Sri Jayewardenepura, Sri Lanka (Grant No: 002/2017)”

Reviewers' comments:

Reviewer's Responses to Questions

**Comments to the Author**

1. Is the manuscript technically sound, and do the data support the conclusions?

Reviewer #1: Yes

Reviewer #2: Yes

2. Has the statistical analysis been performed appropriately and rigorously? 

Reviewer #1: Yes

Reviewer #2: N/A

3. Have the authors made all data underlying the findings in their manuscript fully available?

Reviewer #1: Yes

Reviewer #2: Yes

4. Is the manuscript presented in an intelligible fashion and written in standard English?

Reviewer #1: No

Reviewer #2: Yes

5. Review Comments to the Author

Reviewer #1: Overall, the article presents valuable insights into the needs and concerns of cancer patients in Sri Lanka regarding pain management. It effectively highlights the significant impact of cancer pain on patients' daily lives and emotional well-being, shedding light on the challenges they face. The use of qualitative methods allows for a deep exploration of patients' experiences, adding depth to the discussion. However, there are areas where clarity and consistency could be improved, as mentioned below related to each component;

Abstract- To enhance the abstract, it would be beneficial to include a sentence on cancer prevalence and trends in Sri Lanka,helping readers understand the significance of this study

Introduction-

The introduction section provides relevant information pertinent to the title "Mitigating Cancer Pain: What Else Matters?—A Qualitative Study into the Needs and Concerns of Cancer Patients in Sri Lanka."

It includes important details on the prevalence and impact of cancer pain, both globally and in Sri Lanka, highlights the need for improved pain management, and justifies the use of a qualitative approach. However, there are areas where clarity and emphasis could be enhanced. Here are some comments and suggestions:

1. Cancer Prevalence and Context:

You have effectively included statistics on cancer prevalence globally and in Sri Lanka. Further, better to give more stastics on increasing cancer trend in sri Lanka based on the Cancer Registry data.

However, it would be beneficial to explicitly connect these statistics to the significance of studying cancer pain specifically in Sri Lanka. For instance, "With 29,604 new cases reported in 2020, the growing burden of cancer in Sri Lanka underscores the urgent need to address cancer-related pain comprehensively.".

2. Impact of Cancer Pain:

The discussion on how cancer pain affects various aspects of life is thorough. However, it might be useful to provide more specific examples relevant to the Sri Lankan context if available. This would help in localizing the problem and emphasizing the need for the study.

3. Challenges in Pain Management:

The paragraph discussing the prevalence of untreated cancer pain in Asia compared to Europe and the USA is very relevant. Consider emphasizing the specific challenges faced by Sri Lanka in this context. For example, "In Sri Lanka, these challenges are exacerbated by limited healthcare resources and inadequate pain management facilities."

4. Holistic and Individualized Pain Management:

The introduction effectively argues for holistic and individualized pain management approaches. It might help to briefly mention any existing gaps in Sri Lanka's healthcare system that this study aims to address. For example, "Despite the recognized need for holistic care, Sri Lanka's healthcare system faces significant gaps in implementing such approaches."

5. Justification for Qualitative Approach:

You have justified the use of a qualitative method well. To strengthen this, you could briefly outline what specific insights you hope to gain through qualitative research.

6. Objective of the Study:

The objective of the study is stated, but it could be made more prominent. Consider explicitly stating the research question or hypothesis. For example, "This study seeks to explore the primary needs and concerns of cancer patients in Sri Lanka regarding pain management, aiming to inform more effective and patient-centered interventions."

7. Relevance to Title:

Ensure that the introduction explicitly ties back to the title "Mitigating Cancer Pain: What Else Matters?" by outlining what additional factors (beyond pain relief) are being investigated. For example, "In addition to pain relief, this study will explore the broader psychosocial needs and concerns that impact the quality of life for cancer patients in Sri Lanka."

Comments on the Methods Section:

1. Section 2.1: Study Design

The phrase "A qualitative descriptive study was conducted approach to delve into" is grammatically incorrect. Consider revising it to "A qualitative descriptive study was conducted to delve into."

The sentence "This study adopts a constructivist perspective, acknowledging the subjective nature of cancer experiencing pain..." is slightly awkward. Consider revising it to "This study adopts a constructivist perspective, acknowledging the subjective nature of experiencing cancer pain..."

2. Section 2.2: Study Setting and Participants-

• Mention the specific reasons for selecting Apeksha Hospital as the study setting to ensure that readers, particularly those from outside Sri Lanka, understand its relevance.

• The process of purposeful selection of study participants is not described adequately. A concise yet vivid description of the participants, site, and researcher is essential to provide readers with a comprehensive understanding of the study context.

Ex: It is unclear whether patients with advanced cancer or palliative patients were included. If both curable cancer patients and those with advanced or palliative cancer are included, the interpretation of pain and their needs may vary.

3. Section 2.3: Study Instruments

o Elaborate on the semi-structured interview guide, including whether probing questions were used and what the opening question was / what kind of questions included

o The sentence "The guide’s concepts and question areas were determined based on a comprehensive literature review and subsequently revised by subject specialists" is clear, but could be more concise. Consider: "The guide’s concepts and questions were based on a comprehensive literature review and revised by subject specialists."

4. Section 2.4: Data Collection

o Procedural rigor should be documented by providing information about the researcher’s credentials and previous experience in interviewing and communication to increase reader confidence.

o Since more than one interviewer was used, it is important to describe measures taken to maintain uniformity in data collection, such as training of data collectors.

o Consider breaking up the sentence "The interviewer introduced herself and described the purpose of the interview" into two sentences for clarity: "The interviewer introduced herself and described the purpose of the interview. The informant was assured that the information would be treated with confidentiality..."

5. Section 2.5: Data Analysis

o The description of data analysis lacks detail, particularly regarding the transcription and translation process.

o The data analysis section appears incomplete and need more details on analysis is required;

Was coding performed on the basis of pre-identified themes or just obtained directly from the interviews? Please specify

It is unclear how the investigators in addition to the primary analyst agreed on the initial coding of the raw data; and how the disagreements if any were addressed

• Using "myself" in academic or professional writing is generally discouraged because it can sound informal or overly personal. It's often better to use more formal language, such as referring to oneself as the author, researcher, or principal investigator. This helps maintain a professional tone and clarity in the writing.

o The phrase "The transcripts were generated codes, subcategories, categories, and themes" is unclear. Consider revising it to "Codes, subcategories, categories, and themes were generated from the transcripts."

6. Section 2.6: Trustworthiness

o This section should be revised to provide a more structured and comprehensive overview of the four components of trustworthiness: credibility, confirmability, dependability, and transferability.

o Rather than being scattered throughout the section, each component should be clearly addressed, detailing the specific actions taken by the research team to ensure the overall rigor of the study.

o This restructuring will enhance clarity and facilitate a better understanding of the methodological approach employed to validate the data.

o The phrase "Internal validity was promptly checked on transcripts" could be clarified. Consider specifying what this process entailed.

Overall, your methods section is thorough and well-organized. Making these adjustments will enhance clarity and readability, ensuring your audience can easily follow the methodology.

while your Introduction and methods section is thorough and well-organized, enhancing the grammatical structure and clarity of certain sentences would improve readability. I recommend revising awkward phrases and simplifying complex sentences to ensure clear communication of the article

Results:

Overall, this section offers valuable insights into the lived experiences of cancer patients in Sri Lanka. However, Improvements in grammar and clarity would enhance readability and precision, addressing occasional instances of awkward phrasing and repetitive language. While direct quotes enrich the results, restructuring some sentences for better coherence would be beneficial. Streamlining language and minimizing repetition present opportunities for enhancing the overall quality of the section.

1st Paragaraph –

Age group 51-60: It is not appropriate to refer to this group as the majority since it only represents 33% of the sample. If you want to refer to a majority, consider using broader categories like 31-70 or at least 41-70.

• Include information on whether participants were working, retired, or had stopped working because of cancer, as this is more relevant to their monthly income.

• Time since diagnosis could be categorized further (e.g., 2-3 years, >3 years) to provide a clearer picture of the duration of suffering among included patients.

Table 2. Theme 1: Changing normal lifestyle

• The quote "I cannot cook now. I do not eat the way I want to..." is more indicative of functional limitations rather than an emotional reaction. Consider categorizing it accordingly.

Discussion

• It seems there's a discrepancy between the methods and the discussion regarding who conducted the interviews. While the methods section states that multiple individuals conducted the interviews, the discussion mentions that the same investigator conducted all interviews for data consistency. This inconsistency raises concerns about the accuracy and reliability of the study's methodology. Clarifying this discrepancy is crucial for maintaining transparency and ensuring the credibility of the research findings.

• The discussion could benefit from further exploration of how the study findings contribute to the existing literature on cancer pain management, particularly in the context of Sri Lanka.

• Additionally, expanding the discussion to include potential implications for healthcare practice in Sri Lanka and future research directions would enhance the article's overall contribution to the field.

Reviewer #2: This qualitative study addresses a critical topic that spans individual and social health. It is a well-conducted study, adhering to qualitative methods. However, the following issues need to be addressed:

*Specify whether participants received any rewards for their involvement in the study

*Elaborate on how nonverbal responses were utilized in the subsequent analysis

*Describe the measures taken to ensure the quality of data analysis

*Confirm whether interview transcripts were reviewed by individuals other than the interviewers

*Identify who performed the transcriptions

*Utilize the demographic characteristics collected during the study to enrich the discussion section

*There are a few grammatical mistakes

6. PLOS authors have the option to publish the peer review history of their article (what does this mean?). If published, this will include your full peer review and any attached files.

Reviewer #1: No

Reviewer #2: **Yes: **WDCN Adikaram

---

## [Author Response · Author response to Decision Letter 0]

13 Oct 2024

Comment 1. Please ensure that your manuscript meets PLOS ONE's style requirements, including those for file naming.

Response- Revised the manuscript

Comment 2-Please state what role the funders took in the study

Response-Added the role of the funder

Comment 3. -Please confirm at this time whether or not your submission contains all the raw data required to replicate the results of your study.

Response-Semi-structured Interview guide used to collect data was attached as a supplementary document.

Reviewer #1:

 Abstract:

Comment 1- it would be beneficial to include a sentence on cancer prevalence and trends in Sri Lanka,

Response- Included information on cancer prevalence and trends in Sri Lanka

Comment 2- Introduction: better to give more statistics on the increasing cancer trends in Sri Lanka based on the Cancer Registry data. it would be beneficial to explicitly connect these statistics to the significance of studying cancer pain specifically in Sri Lanka.

Response- given the statistics on the increasing cancer trends in Sri Lanka and connected the statistics to the significance of studying cancer pain in Sri Lanka

Comment 3- Consider emphasizing the specific challenges faced by Sri Lanka in this context

Response- “A clinical audit found suboptimal pain assessment in cancer patients [7], nurses lack autonomy and specialized knowledge in cancer pain management, often working in a task-oriented system that neglects patients' pain needs [8], In Sri Lanka, limited healthcare resources and inadequate pain management facilities exacerbate these challenges.”

Comment 4- It might help to briefly mention any existing gaps in Sri Lanka's healthcare system that this study aims to address. For example, "Despite the recognized need for holistic care, Sri Lanka's healthcare system faces significant gaps in implementing such approaches."

Response-“Despite advancements in medical therapies, there remains a lack of comprehensive understanding of the essential needs and concerns related to cancer pain, particularly in resource-constrained settings like Sri Lanka. The country's healthcare system, while acknowledging the importance of holistic care, struggles with significant gaps in the implementation of such approaches. This is especially crucial given Sri Lanka's ongoing demographic and epidemiological transition, characterized by an aging population and an increasing prevalence of non-communicable diseases.”

Comment 5- you could briefly outline what specific insights you hope to gain through qualitative research.

Response- “In light of these factors, it is vital to emphasize the significance of conducting this study using a qualitative research approach. Qualitative methods provide unique insights into patients' lived experiences, perspectives, and unmet needs, offering a rich understanding of the complexities surrounding cancer pain. By capturing these nuances, the study seeks to identify patient needs and problems of effective pain management and to understand the emotional and psychological impacts of cancer pain in the Sri Lankan context. Ultimately, these insights will inform the development of more effective, patient-centered interventions tailored to the unique needs of cancer patients in Sri Lanka.”

Comment-6- The objective of the study is stated, but it could be made more prominent. Consider explicitly stating the research question or hypothesis. For example, "This study seeks to explore the primary needs and concerns of cancer patients in Sri Lanka regarding pain management, aiming to inform more effective and patient-centered interventions."

Ensure that the introduction explicitly ties back to the title "Mitigating Cancer Pain: What Else Matters?" by outlining what additional factors (beyond pain relief) are being investigated. For example, "In addition to pain relief, this study will explore the broader psychosocial needs and concerns that impact the quality of life for cancer patients in Sri Lanka."

Response -“The study aims to address the challenges associated with cancer pain in Sri Lanka by deepening the understanding of the substantial needs and issues faced by patients. Beyond focusing solely on pain relief, this research will investigate the broader psychosocial concerns that significantly impact the quality of life for cancer patients. These include emotional, psychological, cultural, and social factors that influence patients' experiences and coping mechanisms. By examining these areas, the study seeks to enhance pain recognition and contribute to the development of more effective, patient-centered interventions, ultimately aiming to reduce the overall burden of cancer pain in the country..”

Comments on the Methods Section:

Section 2.1: Study Design

Comment- 

The phrase "A qualitative descriptive study was conducted approach to delve into" is grammatically incorrect. Consider revising it to... 

The sentence "This study adopts a constructivist perspective, acknowledging the subjective nature of cancer experiencing pain..." is slightly awkward. Consider revising it to "..."

Response- Revised as suggested

“A qualitative descriptive study was conducted to delve into the lived experiences of individual needs and major concerns of patients with cancer pain in Sri Lanka. 

This study adopts a constructivist perspective, acknowledging the subjective nature of experiencing cancer pain and seeking to explain and comprehend the many interpretations attributed to these experiences by individuals”

Comment 2- Section 2.2: Study Setting and Participants-

Mention the specific reasons for selecting Apeksha Hospital as the study setting

The process of purposeful selection of study participants is not described adequately

It is unclear whether patients with advanced cancer or palliative patients were included.

Response- Apeksha Hospital is the premier center for cancer care in Sri Lanka. 

The process of purposeful selection of study participants is described

Patients fulfilling inclusion and exclusion criteria were included irrespective of advanced cancer and palliative state.

Comment-3 Section 2.3: Study Instruments

Elaborate on the semi-structured interview guide, including whether probing questions were used and what the opening question was / what kind of questions included

 Response- Elaborated the content of the semi-structured interview guide, including the use of probing questions and what the opening question was / what kind of questions included. The interview guide was submitted as a supplementary file

Comment 4- Section 2.4: Data Collection

Procedural rigor should be documented by providing information about the researcher’s credentials and previous experience in interviewing and communication to increase reader confidence.

Since more than one interviewer was used, it is important to describe measures taken to maintain uniformity in data collection, such as the training of data collectors.

Consider breaking up the sentence "The interviewer introduced herself and described the purpose of the interview" into two sentences for clarity Consider breaking up the sentence

Response- The sentence “The principal investigator (PI), a registered nurse, and an experienced qualitative researcher conducted semi-structured interviews…” was revised for clarity. The PI was the sole data collector. 

“The principal investigator (PI) conducted semi-structured interviews (n=21) until theoretical data saturation was attained with the use of an interview guide. The PI's background in nursing and qualitative research, combined with her previous experience in conducting in-depth interviews and effective communication, ensured the rigorous collection of data and enhanced the reliability of the findings.” 

Revised as suggested

Comment 5- Section 2.5: Data Analysis

The description of data analysis lacks detail, particularly regarding the transcription and translation process.

The data analysis section appears incomplete and need more details on analysis is required;

Was coding performed on the basis of pre-identified themes or just obtained directly from the interviews? Please specify

 Response- Revised the data analysis section with more details. Codes, and themes were obtained from the interviews, there were no pre-identified themes.

Comments- Section 2.6: Trustworthiness

This section should be revised to provide a more structured and comprehensive overview of the four components of trustworthiness: credibility, confirmability, dependability, and transferability.

Response- Revised as suggested

Results:

Comment- 1st Paragaraph –

Include information on whether participants were working, retired, or had stopped working because of cancer

Time since diagnosis could be categorized further (e.g., 2-3 years, >3 years) to provide a clearer picture of the duration of suffering among included patients

 Response- I added information on the current status of the occupation. There was no recorded information on retired or had stopped working because of cancer. 

Time since diagnosis categorized further (e.g., 2-3 years, >3 years).

Comment 8- Table 2. Theme 1: Changing normal lifestyle

Response- The quote "I cannot cook now. I do not eat the way I want to..." is categorized under functional limitations

Discussion

Comment- Discrepancy regarding who conducted the interviews

Response- Clarified in the methods section.

Comment- The discussion could benefit from further exploration of how the study findings contribute to the existing literature on cancer pain management, particularly in the context of Sri Lanka.

Response- Revised and elaborated the first paragraph of the discussion section.

Comment- Expanding the discussion to include potential implications for healthcare practice in Sri Lanka and future research directions

response- The last two paragraphs added including potential implications for healthcare practice in Sri Lanka and future research

Reviewer #2:

Comment- Specify whether participants received any rewards for their involvement in the study

response- The participants did not receive ant reward for their involvement in the study

Comment- Elaborate on how nonverbal responses were utilized in the subsequent analysis

Response- Elaborated on how nonverbal responses were utilized in the subsequent analysis

Comment- Describe the measures taken to ensure the quality of data analysis

 Response- Described under four components of trustworthiness: credibility, confirmability, dependability, and transferability.

Comment- Confirm whether interview transcripts were reviewed by individuals other than the interviewers

 Response-The interviewer (PI) reviewed initially and later the members of the research team involved in reviewing. Later two external qualitative research experts were reviewed

Comment- Identify who performed the transcriptions

Response- Transcriptions were done by the interviewer. 

Comment- Utilize the demographic characteristics collected during the study to enrich the discussion section

Response- Elaborated discussion section including demographic characteristics.

---

## [Decision Letter · Decision Letter 1]

10 Dec 2024

PONE-D-24-13898R1‘‘Mitigating Cancer Pain: What else Matters?”—A Qualitative Study into the Needs and Concerns of Cancer Patients in Sri LankaPLOS ONE

Dear Dr. Edirisinghe,

Thank you for submitting your manuscript to PLOS ONE. After careful consideration, we feel that it has merit but does not fully meet PLOS ONE’s publication criteria as it currently stands. Therefore, we invite you to submit a revised version of the manuscript that addresses the points raised during the review process.

We look forward to receiving your revised manuscript.

Kind regards,

Surangi Jayakody, MBBS, MSc, MD

Academic Editor

PLOS ONE

Journal Requirements:

Reviewers' comments:

Reviewer's Responses to Questions

**Comments to the Author**

1. If the authors have adequately addressed your comments raised in a previous round of review and you feel that this manuscript is now acceptable for publication, you may indicate that here to bypass the “Comments to the Author” section, enter your conflict of interest statement in the “Confidential to Editor” section, and submit your "Accept" recommendation.

Reviewer #1: (No Response)

Reviewer #2: All comments have been addressed

2. Is the manuscript technically sound, and do the data support the conclusions?

Reviewer #1: Yes

Reviewer #2: Yes

3. Has the statistical analysis been performed appropriately and rigorously? 

Reviewer #1: Yes

Reviewer #2: Yes

4. Have the authors made all data underlying the findings in their manuscript fully available?

Reviewer #1: Yes

Reviewer #2: (No Response)

5. Is the manuscript presented in an intelligible fashion and written in standard English?

Reviewer #1: Yes

Reviewer #2: Yes

6. Review Comments to the Author

Reviewer #1: I appreciate the authors' effort in addressing the previous review comments and improving the manuscript. The study provides valuable insights, and the revisions made so far have strengthened the overall quality. However, there are still a few areas where further refinement could enhance clarity and precision, as detailed below:

Abstract:

• Clarity and Conciseness: The abstract is mostly clear, though readability could be improved. For example, “patients eighteen years older with cancer-related pain were recruited” could be rephrased to clarify that participants were 18 years

or older. Overall, the abstract is informative but could benefit from minor refinements to enhance clarity and conciseness.

Method of Data Collection: as the method of data collection the word “In-depth interviews” are not clearly mentioned in the abstract and methodology sections.

Administrative Clearance: There is no mention of administrative clearance anywhere in the article.

Specific Language Suggestions:

• Sentence 231 : The term 'multiple sources of data' is misleading, as only patient interviews were conducted. 'Multiple sources' would be appropriate if data were also obtained from healthcare staff, family members, caregivers, or other sources. Please revise to accurately reflect the data sources used in this study

• Sentence 262: Consider replacing “salary” with “income” for accuracy.

• Sentence 478: The use of “triangulation” is unclear; if triangulation was not done, please remove this term.

Results and Discussion: Aim for a more concise, focused presentation of the results and discussion sections, avoiding repetition.

Reviewer #2: (No Response)

7. PLOS authors have the option to publish the peer review history of their article (what does this mean?). If published, this will include your full peer review and any attached files.

Reviewer #1: No

Reviewer #2: **Yes: **WDCN Adikaram

---

## [Author Response · Author response to Decision Letter 1]

12 Dec 2024

Dr. NP Edirisinghe,

Senior Lecturer

Department of Fundamental Nursing, 

Faculty of Nursing, University of Colombo, Sri Lanka

12.12.2024

Academic Editor and Reviewers

PLOS ONE

Submission of revised version 02 of original research paper to PLOS ONE

I would like to express my sincere gratitude for your valuable feedback and constructive comments on my manuscript.

The revisions suggested are tabulated as follows.

Journal Requirements:

 Comment Response

1. Please review your reference list to ensure that it is complete and correct. If you have cited papers that have been retracted, please include the rationale for doing so in the manuscript text, or remove these references and replace them with relevant current references. Any changes to the reference list should be mentioned in the rebuttal letter that accompanies your revised manuscript. If you need to cite a retracted article, indicate the article’s retracted status in the References list and also include a citation and full reference for the retraction notice. Reviewed the reference list and revised the 29th reference.

Reviewer #1:

1. abstract is informative but could benefit from minor refinements to enhance clarity and conciseness Revised the suggested sentence

2. as the method of data collection the word “In-depth interviews” are not clearly mentioned in the abstract and methodology section The data collection method was not ‘In-depth interviews’, it was ‘semi structured interviews. It was mentioned in the abstract and the methodology section.

3. Administrative Clearance: There is no mention of administrative clearance anywhere in the article. The sentence on ‘Administrative Clearance’ mentioned under the section 2.4.

4. Sentence 231 : The term 'multiple sources of data' is misleading, Please revise to accurately reflect the data sources used in this study Revised enhancing the clarity.

5. Sentence 262: Consider replacing “salary” with “income” for accuracy. Replaced “salary” with “income”

6. Sentence 478: The use of “triangulation” is unclear; if triangulation was not done, please remove this term. Revised as per the suggestion

7. Results and Discussion: Aim for a more concise, focused presentation of the results and discussion sections, avoiding repetition. Some of the repeating contents were removed. Some of the repetition in the manuscript is intentional and serves to emphasize the multifaceted nature of cancer pain, ensuring clarity, depth, and contextual relevance across themes. It reflects recurring patterns in participants' experiences, supporting the validity and trustworthiness of the findings while maintaining their voices at the center of the analysis.

Reviewer #2: No comments

Thank you for your consideration!

Sincerely,

Nirosha Edirisinghe, PhD

Corresponding author

---

## [Editor Report · Decision Letter 2]

6 Jan 2025

‘‘Mitigating Cancer Pain: What else Matters?”—A Qualitative Study into the Needs and Concerns of Cancer Patients in Sri Lanka

PONE-D-24-13898R2

Dear Dr. Nirosha,

We’re pleased to inform you that your manuscript has been judged scientifically suitable for publication and will be formally accepted for publication once it meets all outstanding technical requirements.

Kind regards,

Surangi Jayakody, MBBS, MSc, MD

Academic Editor

PLOS ONE

---

## [Editor Report · Acceptance letter]

10 Jan 2025

PONE-D-24-13898R2 

PLOS ONE

Dear Dr. Edirisinghe, 

I'm pleased to inform you that your manuscript has been deemed suitable for publication in PLOS ONE. Congratulations! Your manuscript is now being handed over to our production team.

Kind regards, 

on behalf of

Dr Surangi Jayakody 

Academic Editor

PLOS ONE
